# A multimodal transformer-based tool for automatic generation of concreteness ratings across languages
Viktor Kewenig ✉, Jeremy I. Skipper & Gabriella Vigliocco

We present an automated method for generating concreteness ratings that achieves beyond human-level reliability across multiple languages and expression types. Our approach combines multimodal transformers with emotion-finetuned language models and achieves correlations of 0.93 for single British words and 0.85 for multiword expressions with existing corpora of human raters. We demonstrate general applicability through successful cross-lingual generalization to an entirely unseen corpus of Estonian single- and multi-word expressions (N = 35,979), achieved via automated language detection and translation. By leveraging both visual and emotional information in context-aware language embeddings, our method effectively captures the full spectrum from concrete to abstract concepts. Our automated system offers a context sensitive, reliable alternative to traditional human ratings, eliminating the need for time-consuming and costly human rating collection. We provide an easy to access web-based interface for research to use our tool under concreteness.eu.

Concreteness evaluates the degree to which the concept denoted by a word refers to a perceptible entity. The variable first came to the foreground of Psychology in Paivio's dual-coding theory[1,2]. According to this theory, human cognition operates with two distinct classes of mental representation: (1) verbal representation that encode linguistic statistical regularities on the one hand and (2) mental images that encode perceived experiences. Opposed to this view is the idea that all content of human cognition emerges in some way from mental "simulations" of experience[3]. Since then, more recent views have argued that both types of mental representations interact with each other, placing more or less emphasis on encoding of linguistic distributive patterns[4] or contextualized, real-world experience[5].

Underlying this ongoing theoretical debate is a continuous stream of behavioral and neurobiological work trying to decipher the mechanisms of human conceptual processing[6–8]. A lot of this highly influential work has made use of concreteness ratings, collected in large-scale experiments with human participants. One such corpus created by Brysbaert et al.[9] employed over 4000 human raters and contains concreteness ratings for almost 40,000 words, which are amply used in this line of research. For example, one developmental study used these ratings to examine how abstract world knowledge develops in children[10]. Another, neurobiological study dis-associating the cortical networks processing verbs and nouns used the ratings to match verbs and nouns for concreteness (a potentially confounding variable)[11]. Related work has used these ratings to examine contextual modulation of abstract and concrete encodings in the brain during natur-alistic processing[12]. Finally, a more computationally focused use case is

Cunha et al. who created an algorithm for automatic emoji creation (given a textual prompt) and used the ratings to evaluate whether more concrete text prompts were easier to "emojize". Overall, the corpus generated by Brysbaert et al. has been mentioned in more than 2000 peer reviewed journal papers to this day and it is therefore safe to say that concreteness ratings have become an important tool in Cognitive Science, Neuroscience, Psycholinguistics and even computational applications. Considerable efforts have therefore gone into collecting human concreteness ratings (e.g., see Brysbaert et al.[9] and Muraki et al.[13]).

There are at least three issues with these and related corpora of human concreteness ratings. For one, they include only a limited number of words. For example, the 40,000 English words covered in Brysbaert et al.[9] do not even cover a quarter of the total of words currently mentioned in the Oxford Dictionary. Furthermore, this corpus excludes many compund- or multi-expression words. Muraki et al.[13] did go through the effort of collecting multiword ratings, yet compared to the countless possible combinations of multi-expression words, their number of collected words still appears small. Second, collecting human ratings is costly and time-intensive. Brysbaert et al.[9] required an estimated 17,000 h of human labor. Another important issue with both single-word and multiword databases is that they are rated out of context, yet words are polysemous and known to have different meanings in different contexts. It would be impossible for any number of humans to rate all of these. Finally, (and likely due to the costliness of collecting data) only a few comprehensive concreteness ratings exist for languages aside English and Estonian, all of which are either small[14–18] or

Experimental Psychology, University College London, London, UK. ✉e-mail: ucjuvnk@ucl.ac.uk

focused on Western-European languages[19]. It is a well-known issue that a lot of research in psychology is focused on WEIRD societies (Western, educated, industrialized, rich and democratic). Therefore, making concreteness ratings easily available in other languages and for research in other types of societies is a desideratum.

For these reasons, an automated method for generating on-demand concreteness ratings, which could easily transfer from English into other languages, is bound to be a useful tool for current and future research in cognitive science, neuroscience and psycholinguistics. Additionally, the notion of concreteness is gaining significance in semantic-oriented natural language processing tasks. For example, Turney et al.[20] present a supervised model that exploits concreteness to correctly classify 79% of adjective-noun pairs as having literal or non-literal meaning. Tsvetkov et al.[21] exploit both the notions of concreteness and imageability to perform metaphor detection on subject-verb-object and adjective-noun relations, correctly classifying 82% and 86% instances, respectively. Because of the aforementioned benefits of automatically generating reliable concreteness ratings, a few methods already exist.

## Previous work

First, Kwong[22] used a dependency parsing procedure defined by Johansson and Nugues[23] on wordnet[24]. The underlying intuition was to exploit the regularity exhibited in definitions in the sense that the more concrete a word, the more conveniently and convincingly it can be explained with reference to its superordinate concept and distinguishing features. However, this hypothesis lacks independent empirical justification. The results presented by Kwong[22] are insufficient because they are limited to a set of 100 nouns. Indeed, the dependency parsing procedure presented is inherently limited to nouns because it takes "genus" as a defining feature of concreteness. Since only nouns have genera, this method cannot extrapolate to other types of words.

A second method employed by both Ljubešić et al.[25] as well as Charbonnier and Wartena[26], is to use word embeddings from a large model based on global distributional co-occurence statistics for training a regression model to predict concreteness scores. Charbonnier and Wartena use "fast text"[27] and "GoogleNews"[28] embeddings to train a support vector classifier on the Brysbaert et al.[9] corpus discussed above. They evaluate their method on a little less than 4000 English words. As part of their predictive features, they selected frequent suffixes to identify word-types. Their main reason for this is their assumption that nouns are mostly concrete. This assumption is unwarranted, as indeed a lot of nouns are abstract[29]. Because their test set includes mostly nouns, including this feature likely boosted performance on their evaluation. Therefore, it is questionable to what extent this methodology can extrapolate to other word-types.

A third methodology employed including transfer to a foreign language is employed by Ljubešić et al.[25], who use the "fast text" model architecture trained on Wikipedia dumps with embedding spaces aligned between languages. They combine this approach with a linear transformation learned via singular value decomposition[30] on a bilingual dictionary (Croatian and English) of 500 out of the 1000 most frequent English words, obtained via the Google Translate API. They also train a support vector machine on two sets of training corpora. For their "within-language" experiment, they use a training corpus of almost 45,000 english words and evaluate on 3 separate data sets consisting of concreteness ratings for around 3000 words each. The main point of this methodology is not to report state-of-the-art performance on matching human concreteness ratings, however. Rather, they conduct the within-language experiment because their use of linear transformation on bilingual data allows them to also conduct an "across-language" experiment, which adds to the training data a corpus of 3000 Croation words (which they themselves created in an online experiment for the purpose of this study). Their results suggest that their linear transformation successfully transfers concreteness ratings from one language to the other with a loss of roughly 15% accuracy.

While the possibility of easy and fairly lossless transfer of concreteness ratings between languages using word embeddings is certainly intriguing,

the use of embeddings based on co-occurence statistics has two important drawbacks, which may well have a negative impact on performance. It is a well known fact in the cognitive and neuroscience literature on conceptual processing that abstract words vary significantly in meaning, depending on their linguistic context[31,32]. However, embeddings from models based on distributional co-occurence statistics are static and cannot account for contextual variance. Thus, embeddings for abstract words in particular will likely not be a good feature for any classifier (regression-based or other). This latter point is also illustrated by results in Hill and Korhonen[33] who used an approach based on distributional co-occurence statistics to predict concreteness ratings and found that this was significantly impaired for abstract words.

A fourth interesting approach for predicting concreteness ratings is presented in Haagsma and Bjerva[34], who, rather than relying on the richness of embeddings from modern language models, use the linguistic concept of "selectional preference" as a feature for predicting concreteness. Selectional preferences are defined as the tendency of predicates to impose semantic restrictions on the realizations of their complements, i.e., co-occurrence in a syntactic predicate-argument relationship[35]. For example, the word "eat" places the selectional preference of an edible thing as direct object (if not used metaphorically, sentences that violate this principle such as "I eat my education" are nonsensical). Selectional preference may be a good feature for predicting concreteness, because abstract words are likely adjacent to less selectionally preferred predicates (for example: "think about x"), where x could be almost anything. This intuition is closely connected to the fact, discussed above, that abstract concepts are more variable than concrete concepts. However, this method suffers from a similar flaw as distribution-based approaches reviewed above. Selectional preference of a word does not change based on context and therefore inherits the problems from static embedding methods, namely the inability to capture the semantic variability of abstract words in different contexts. Indeed, when presenting their results, Tater et al. admit that the accuracy of their predicted concreteness ratings drops significantly for abstract words.

The best-performing methods for automatically generating concreteness ratings so far Ljubešić et al.[25] achieved correlations of 0.72 with human ratings when tested on an extended set of several thousand English words, with even poorer performance ($r = 0.61$) for cross-lingual transfer. Selectional preference approaches[34] achieve even lower correlations of 0.68. Finally, while Charbonnier and Wartena[26] fared better correlation-wise, reaching a ceiling of around 0.90, their approach was limited to nouns only.

We submit that these shortcomings stem from a fundamental limitation of current approaches in failing to incorporate key insights from cognitive science about how different types of words are grounded in human experience. Research has shown a crucial distinction: while concrete words are primarily grounded in sensory-perceptual experience, abstract words tend to be grounded in emotional and introspective experience[8,36] (but see ref. 37 for a contrasting account). For example, the meaning of concrete words like "table" or "pencil" is largely determined by their visual-sensory properties, while abstract concepts like "freedom" or "anxiety" derive their meaning primarily from emotional and experiential associations. Current methods based on distributional statistics or selectional preferences cannot capture either type of grounding - they neither incorporate the visual-sensory information crucial for concrete concepts nor the emotional content essential for abstract ones. This disconnect from how humans actually represent and process different types of concepts significantly limits these methods' ability to capture the full semantic depth of concreteness ratings. An optimal approach would need to incorporate both types of information, creating an embedding space that captures both the visual-sensory grounding of concrete words and the emotional grounding of abstract ones.

## Our approach

To overcome the limitations of current methods, we leverage recent advances in multimodal, transformer-based architectures that offer richer, context-sensitive language representations. Transformer models learn to predict each subsequent token in a sequence by attending selectively to other

tokens, enabling them to dynamically incorporate linguistic context into their embeddings[38]. Through this attention mechanism, transformers can, for example, differentiate the meaning of homographs like "bank" across distinct contexts-whether it refers to a financial institution or a scheduled public holiday-an ability that static embedding models lack[39–41].

Modern multimodal transformers push these capabilities further by integrating both textual and visual information into a unified embedding space[42]. Such models are trained on large-scale image-text datasets[43], allowing them to capture not only distributional statistics from language but also visual-semantic relationships, including object appearance and contextual cues. Empirical evidence indicates that incorporating visual features enhances the predictive accuracy of semantic models, particularly for concepts tied closely to perceptible attributes[33]. These multimodal embeddings thus hold substantial promise for capturing the concreteness continuum-from purely abstract notions to vividly tangible concepts.

Yet, an additional dimension-emotional grounding-is critical for modeling abstract concepts effectively. While concrete words often derive meaning from visual-perceptual features, abstract words may rely more on emotional and affective associations[8,36]. To incorporate this dimension, we build upon newly available large-scale datasets of emotionally annotated images[44]. By fine-tuning a multimodal model like CLIP (Contrastive Language Image Pretraining) on this affective data, we can enrich the model's representation space to reflect both the perceptual grounding of concrete words and the affective information essential for abstract ones[45].

In sum, our approach directly addresses the shortcomings of static, distribution-based methods by combining four key elements: (1) transformer-based contextual embeddings for dynamic, context-sensitive semantic representations; (2) multimodal training to integrate visual information that underpins concreteness; (3) emotion-aware finetuning to capture the affective content often central to abstract meanings; and (4) zero-shot generalization to new languages and expression types. As shown in the following sections, this integrated, context-sensitive, multimodal, and emotionally grounded framework yields high predictive accuracy, with correlations exceeding 0.90 for single English words, 0.85 for English multi-word expressions, and a robust 0.68 for an entirely distinct language (Estonian, $r = 0.80$ after post-hoc item exclusion). These results mark a substantial step forward in automated concreteness rating generation

## Methods and data

We developed an approach to automatically generate concreteness ratings by leveraging recent advances in multimodal transformers and emotion-aware language models. The tool can be freely accessed under concreteness. eu Our methodology comprises four main components: (1) a dual-embedding model architecture that combines visual-linguistic and emotional information, (2) a training procedure utilizing large-scale human-annotated datasets, (3) comprehensive evaluation metrics to assess prediction accuracy, and (4) a general prediction system capable of generating reliable concreteness ratings across both single words and multi-word expressions in multiple languages.

Figure 1 represents the pipeline for our approach, which is detailed in section "Model Architecture" below.

## Model architecture

We developed a model that integrates multimodal transformers with emotion-aware language models through a dual-embedding approach, aiming to enhance the prediction of concreteness ratings for words. The architecture comprises three main components: a base visual-language model, an emotion-aware language model, and a deep regressor (explained below) that combines embeddings from both models.

The base visual-language model employed is the Contrastive Language-Image Pre-training (CLIP) model[42], which is designed to learn joint representations of images and text by aligning them in a shared embedding space. Specifically, CLIP comprises two main components: a transformer-based text encoder, which processes textual inputs using a standard transformer architecture, and a vision encoder, which can either be

a ResNet or a Vision Transformer (ViT). In our implementation, we used the ViT-B/32 variant for the vision encoder, which utilizes a Vision Transformer architecture with an image patch size of 32. CLIP was trained on ~400 million image-text pairs sourced primarily from publicly available, web-scale datasets[42].

To incorporate emotional context into our model, we fine-tuned CLIP on the Affection dataset, a collection compiled for this study that consists of 85,007 emotionally annotated images and 526,749 emotional reactions from 6283 unique participants. The dataset includes images sourced from various public repositories, such as the International Affective Picture System (IAPS)[46], and each image is associated with one or more emotional labels. The fine-tuning process involved presenting the text encoder of CLIP with the emotional labels as text inputs and the corresponding images, employing a contrastive loss similar to the original CLIP training but focused on aligning images with their emotional descriptions. This emotion-aware variant of CLIP allows the model to capture emotional information in language and imagery. We compared performance metrics for CLIP, CLIP-Emotion and a combination of both (see Results) and found that combining both embeddings enhanced the model's ability to capture concreteness perception.

The deep regressor is a neural network designed to combine the embeddings from both the base CLIP model and the emotion-aware CLIP model to predict the concreteness of words. The input layer receives the concatenated embeddings, resulting in a combined feature vector of 1024 dimensions (512 from each model). This is followed by two fully connected hidden layers with 128 and 64 units, respectively, each using ReLU activation functions to capture complex nonlinear relationships. Dropout regularization with a rate of 0.2 is applied after each hidden layer to prevent overfitting. The output layer consists of a single unit with a linear activation function that predicts the concreteness score for a given word.

## Training procedure

We used the concreteness ratings dataset compiled by Brysbaert et al.[9], which provides human-annotated concreteness ratings for 37,058 English words and 2896 two-word expressions. Each word was rated on a 5-point Likert scale, where 1 indicates high abstractness and 5 indicates high concreteness. To prepare the dataset for training, we performed data cleaning to remove entries with missing values or non-standard characters, while homographs and polysemous words were retained to maintain linguistic diversity.

The dataset was randomly split into a training set and a test set, allocating ($N = 36,058$) for training and ($N = 1000$) for testing. Stratified sampling was used to maintain the distribution of concreteness ratings in both sets, ensuring that both concrete words (ratings $\geq 2.88$) and abstract words (ratings $< 2.88$) were proportionally represented.

For each word in the dataset, we generated embeddings using both the base CLIP model and the emotion-aware CLIP model. The words were preprocessed by converting them to lowercase and removing any leading or trailing whitespace. Two-word expressions were concatenated with an underscore (e.g., "ice cream" becomes "ice_cream") to be treated as single tokens. The preprocessed words were tokenized using CLIP's byte-pair encoding tokenizer and passed through CLIP's text transformer to obtain 512-dimensional embeddings from each model. These embeddings were concatenated to form a 1024-dimensional feature vector for each word. We applied StandardScaler normalization to the combined embeddings, centering the data to have zero mean and unit variance, which aids in the convergence of the neural network.

The deep regressor was trained using a Mean Squared Error (MSE) loss function to measure the discrepancy between the predicted and actual concreteness scores. We used the Adam optimizer[47] with an initial learning rate of $5 \times 10^{-4}$ and employed a learning rate scheduler that reduced the learning rate by a factor of 0.1 if the validation loss did not improve for three consecutive epochs. Training was conducted with a batch size of 50, and early stopping was implemented to halt training if the validation loss did not decrease for five consecutive epochs. A validation set comprising 10% of the

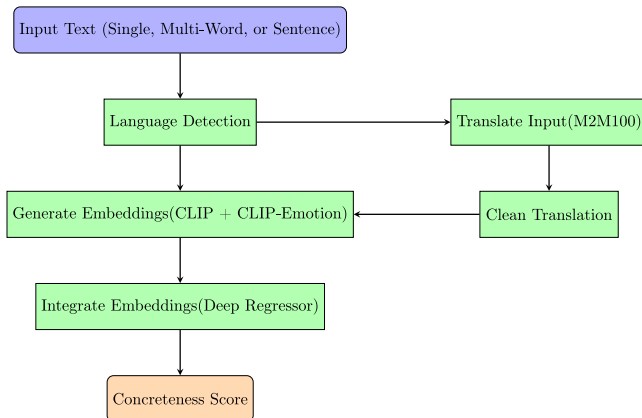

**Fig. 1 | System architecture for generating concreteness ratings.** The process begins with input text (single, multi-word or sentence) undergoing language detection. For non-English inputs, a cross-lingual path usses translation and cleaning steps before embeddings are generated. Embeddings from CLIP and CLIP-Emotion are integrated using a deep regressor to produce the final concreteness score. If the input is a sentence, a concreteness rating will be generated for each word in the input.

training data was used to monitor the model's performance during training. In addition to dropout, L2 regularization with a weight decay of $1 \times 10^{-5}$ was applied to the weights of the network to prevent overfitting.

### Evaluation

We assessed the model's performance using several evaluation metrics on the held-out test set. The Pearson correlation coefficient ($r$) was calculated to measure the linear correlation between the predicted and actual concreteness scores, providing insight into the strength and direction of the relationship. The coefficient of determination ($R^2$) was used to indicate the proportion of variance in the concreteness ratings that is predictable from the model, offering a measure of explanatory power. Mean Absolute Error (MAE) and Root Mean Square Error (RMSE) were computed to quantify the average magnitude of the prediction errors, with MAE representing the average absolute differences and RMSE giving more weight to larger errors due to the squaring of differences. Statistical significance was evaluated using two-tailed t-tests, with a threshold of $p < 0.001$ considered significant, to determine the likelihood that the observed correlations occurred by chance.

### General prediction system

To create a general prediction system capable of handling both single words and multi-word expressions, we developed a combined architecture that integrates our previously trained models. This system merges two specialized models: the single-word model trained on the Brysbaert corpus[9], comprising 37,058 words, and the multi-word model trained on the Muraki corpus[13], which includes 62,000 multi-word expressions. By integrating these models, the system can process and predict concreteness ratings across a diverse range of linguistic inputs.

The system employs a routing mechanism to classify expressions and direct them to the appropriate model. Initially, expressions are tokenized using whitespace delimitation to separate individual words. Secondary tokenization accounts for punctuation and special characters, ensuring accurate determination of expression length. Expressions are classified based on the number of tokens: single tokens are routed to the single-word model, while multiple tokens are directed to the multi-word model. Compound words, such as those containing hyphens, are treated as single words to preserve their semantic integrity. Dynamic model loading is implemented with GPU memory optimization in mind, allowing for efficient utilization of computational resources. The system supports parallel prediction for batched inputs, enhancing processing speed when handling large datasets.

To extend the system's applicability to non-English inputs, we incorporated a cross-lingual component using a multi-stage translation process. The first stage involves language detection using the `langdetect` library, with fallback options for ambiguous cases. A confidence threshold is set to handle uncertain detections, and language-specific preprocessing rules are applied to account for linguistic variations. This means our method is fundamentally monolingual in its core embedding representations, with multilingual applicability achieved through this translation step.

The translation process employs the M2M100 transformer model[48], a multilingual machine translation model with 1.2 billion parameters capable of translating between 100 languages without relying on English as an intermediary. Source language-specific tokenization is applied to prepare the text for translation. We use beam search decoding with a beam size of five to generate the most probable translations, setting a maximum sequence length of 128 tokens to accommodate lengthy inputs. Forced English output token initialization guides the model toward producing English translations.

After translation, a cleaning pipeline ensures the translated text is suitable for further processing. This includes the removal of articles ("a", "an", "the"), normalization of punctuation, and case normalization to maintain consistency. Articles ("a", "an", "the") are uniformly removed from both original and translated text inputs to ensure consistency across languages and prevent translation-induced biases in concreteness predictions. Special handling is implemented for ellipses by truncating trailing periods, and hyphenation is normalized to standard forms. Conjunctions are removed to simplify complex expressions, and in cases where multiple translation candidates are generated, a resolution strategy selects the most semantically appropriate option.

Efficient batch processing is crucial for handling large volumes of input data. Memory management techniques are employed, such as dynamic batch size adjustment based on input length and GPU memory monitoring, to optimize resource utilization. Gradient accumulation is used for large batches to prevent out-of-memory errors, and automatic batch size reduction is triggered if memory constraints are detected. The processing pipeline operates with a batch size of 50 expressions per forward pass, allowing for parallel embedding generation. Synchronized model switching ensures that the appropriate model is loaded and used for each batch segment. Embeddings for repeated expressions are cached to avoid redundant computations, enhancing overall efficiency.

To ensure robustness, the system incorporates comprehensive error-handling mechanisms for both translation and prediction processes. In the translation component, edge cases such as empty translations are addressed through fallback strategies, and unknown languages are handled by defaulting to English processing. Non-ASCII characters are managed with appropriate encoding, and translation verification checks are conducted, including length ratio validation, semantic similarity checking, and character set validation to ensure the integrity of translations.

In the prediction component, out-of-distribution inputs are detected, and confidence thresholding assesses the reliability of predictions. The single-word and multi-word models operate independently and process inputs in parallel. If disagreements arise between model predictions (e.g., for compound words or ambiguous inputs), an ensemble-based disagreement resolution strategy is employed, selecting the prediction associated with the higher model confidence. The system handles very long expressions exceeding ten words, rare or unknown words, mixed-language expressions, and inputs containing special characters and symbols by implementing specific processing rules and fallback mechanisms.

Because we will open-source this project, we added several good practices for easy hands-on coding by interested researchers. For one, recovery procedures are in place to manage errors gracefully. The system offers options for graceful degradation, such as providing partial predictions when full processing is not possible. Fallback prediction strategies, like defaulting to average concreteness scores, are used when inputs cannot be processed reliably. Additionally, a warning and logging system records instances of errors and edge cases, allowing for manual review and future system improvements. Inputs that present significant challenges are flagged

**Fig. 2 | Model predictions vs. true ratings for English expressions.** The strong correlation ($r = 0.93$) indicates high prediction accuracy.

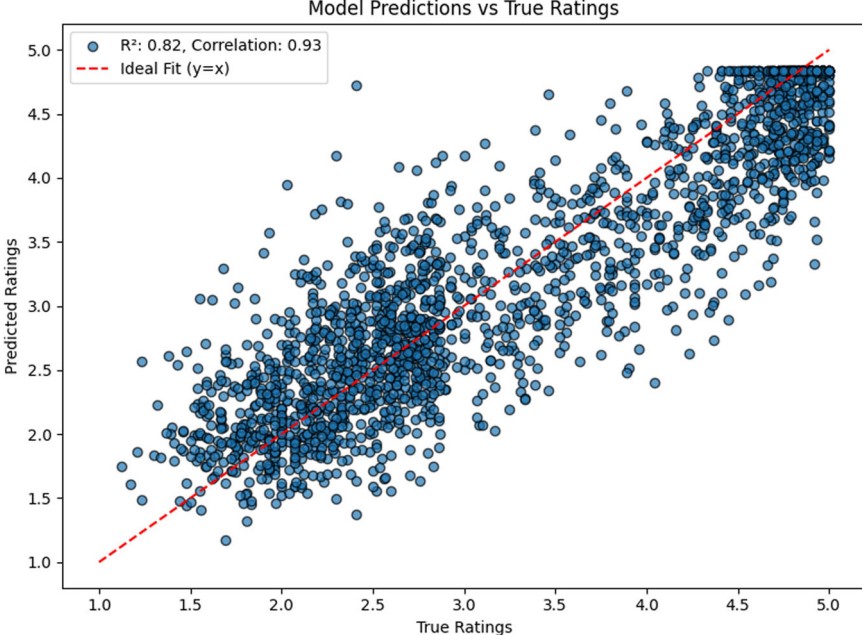

for manual review to ensure accuracy and reliability. Finally, the system's modular design allows for easy updates and improvements to individual components without necessarily affecting the overall architecture.

### Implementation details

All models were implemented using the PyTorch framework[49] and trained with NVIDIA GPU acceleration to expedite computation. Training was conducted on Google Colab with an NVIDIA Tesla A100 GPU with 32 GB of memory and CUDA version 11.2, significantly reducing computation time compared to CPU-based training. We used PyTorch version 1.8.1 and Python 3.8.5, and accessed the CLIP model and tokenizer via the OpenAI CLIP repository. Random seeds were set to 42 for NumPy and PyTorch to ensure reproducibility. Batch processing with shuffling was enabled during training to enhance memory efficiency and model robustness by ensuring that each mini-batch was a representative sample of the dataset. The codebase, including data preprocessing scripts, model architecture definitions, and training routines, is available the following Github repository. The complete training process, including fine-tuning the emotion-aware CLIP model and training the deep regressor, required approximately four hours to converge.

### Validation

We evaluated our method's performance on three distinct datasets: a held-out test set of 1000 words from an English single-word corpus, a held-out test set of 1000 words from an English multi-word corpus, and a complete (unseen) Estonian corpus ($N = 35,979$)[50]. Statistical analyses revealed significant correlations across all evaluations ($p < 0.001$).

To contextualize our model's performance, we first assessed human inter-rater reliability in our datasets. With access to individual participant-level data, we computed split-half reliability of Brysbaert et al.[9] by randomly dividing participants into two groups over 100 iterations. For each iteration, mean ratings per word were computed for both groups, and Pearson correlations were calculated for the common words. The average correlation was then adjusted using the Spearman-Brown correction, yielding a corrected reliability estimate of 0.9077. This result shows that our method indeed surpasses human-based rating reliability. Individual-level participant data could not be obtained for Muraki et al.[13] and Proos and Aigro[50], so reliability was calculated with intraclass correlation coefficients (ICC) using a multi-level model with a random intercept and the expression as a fixed effect. These yielded an estimate of 0.84 for the multi-word expression

corpus and a reliability of 0.81 for the corpus of Estonian concreteness ratings.

### English single words and expressions

For English, our model achieved strong correlations between predicted and true ratings, reaching $R^2 = 0.82$ ($r = 0.93$, 95% CI [0.90, 0.95]) on the held-out test set ($N = 1000$). The scatter plot reveals a tight distribution around the ideal prediction line (Fig. 1), with particularly strong performance in the mid-range of concreteness ratings (2.5–4.0).

The density plot (Fig. 2) demonstrates that our model successfully captures the bimodal distribution characteristic of human concreteness ratings, with peaks around ratings of 2.5 and 4.5, closely matching the distribution of true ratings.

Error analysis reveals a mean absolute error (MAE) of 0.45 points on the 5-point scale, with slightly better performance for concrete words (MAE = 0.41) compared to abstract words (MAE = 0.48). Performance was consistent across parts of speech, with comparable correlations for nouns ($r = 0.90$), verbs ($r = 0.88$), and adjectives ($r = 0.80$).

### Multi-word expression performance

We further evaluated our model's performance on multi-word expressions, testing on a held-out set of English multi-word expressions (N=1000). The model maintained strong performance, achieving $R^2 = 0.71$ ($r = 0.85$, 95% CI [0.84, 0.86]). The scatter plot (Fig. 3) shows a clear linear relationship between predicted and true ratings, although with slightly more variance than single-word predictions.

The slight decrease in performance compared to single words ($r = 0.93$ vs $r = 0.85$) is expected given the increased complexity of multi-word expressions, where meaning often emerges from the interaction between constituent words. Nevertheless, the strong correlation demonstrates that our model successfully captures these compositional effects on perceived concreteness.

The density plot (Fig. 4) again captures the characteristic bimodal distribution of concreteness ratings, with peaks at ~2.0 and 4.5, closely matching the distribution in the true ratings.

### Cross-lingual transfer

To test general applicability, we evaluated the model on an Estonian corpus ($N = 35,979$). The overall model performance was robust with $R^2 = 0.56$ ($r = 0.68$). After excluding cases that exhibited extremely high

**Fig. 3 |** Density distribution of predicted vs. true ratings for English expressions, showing successful capture of the bimodal distribution characteristic of concreteness ratings.

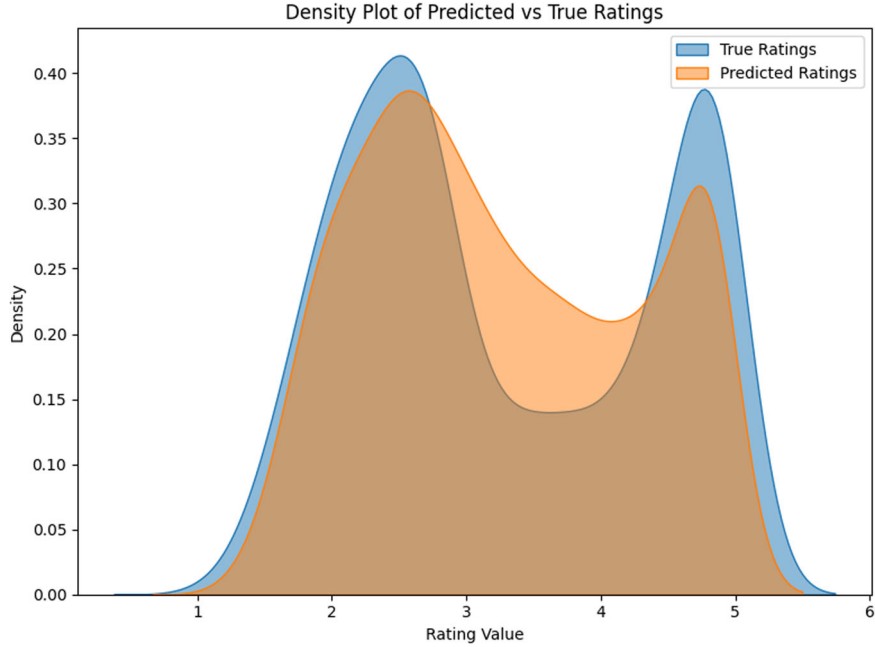

**Fig. 4 | Model predictions vs. true ratings for English multi-word expressions.** The strong correlation (*r* = 0.85) demonstrates robust generalization to complex expressions.

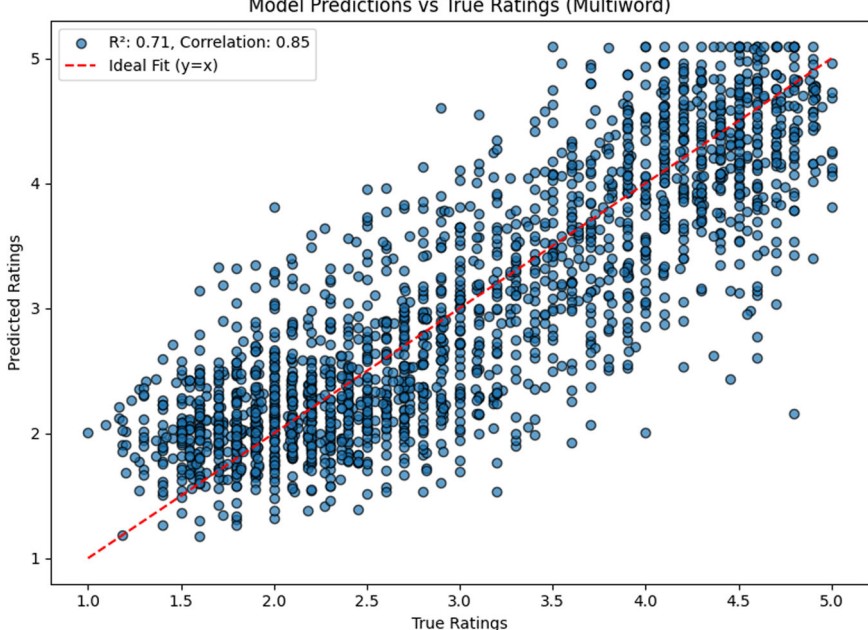

deviation between human raters (beyond 3 standard deviations from the mean), the model performance went up to $R^2 = 0.74$ (*r* = 0.80, 95% CI [0.86, 0.88]). The hexagonal heatmap (Fig. 3) illustrates the density of the remaining 31,470 predictions, with darker blue indicating higher concentration of data points. The tight clustering along the ideal prediction line demonstrates strong cross-lingual transfer, particularly for ratings between 2.0 and 4.5.

This cross-lingual error analysis shows an MAE of 0.52, only marginally higher than English performance despite the language barrier. Systematic error analysis revealed a slight tendency to overpredict ratings for very abstract words (ratings < 2.0, mean bias = +0.3) and underpredict for highly concrete words (ratings > 4.5, mean bias = −0.2). This pattern was consistent across both languages (English and Estonian), suggesting a general model characteristic rather than language-specific bias.

The English test set (*N* = 1000) showed a balanced distribution of concreteness ratings (mean = 3.2, SD = 1.1), with 42% concrete words (rating > 3.5), 38% abstract words (rating < 2.5), and 20% intermediate. The Estonian dataset (*N* = 35,979) showed a similar distribution (mean = 3.3, SD = 1.2), with 44% concrete, 35% abstract, and 21% intermediate words, suggesting comparable conceptual distributions across languages. Notably, the density distributions for both single- and multi-word expressions (Figs. 2, 4) show that our model successfully captures the characteristic bimodal pattern of concreteness ratings, suggesting that it has learned fundamental features of conceptual concreteness that generalize across languages (Fig. 5). The strong performance on both English (*r* = 0.93) and Estonian (*r* = 0.68 and *r* = 0.80 after post-hoc item exclusion) datasets indicates that our method provides reliable automated concreteness ratings that are language-agnostic. However, while the English training data show a

**Fig. 5 |** Density distribution of predicted vs. true ratings for multi-word expressions, showing successful capture of the bimodal distribution characteristic of concreteness ratings.

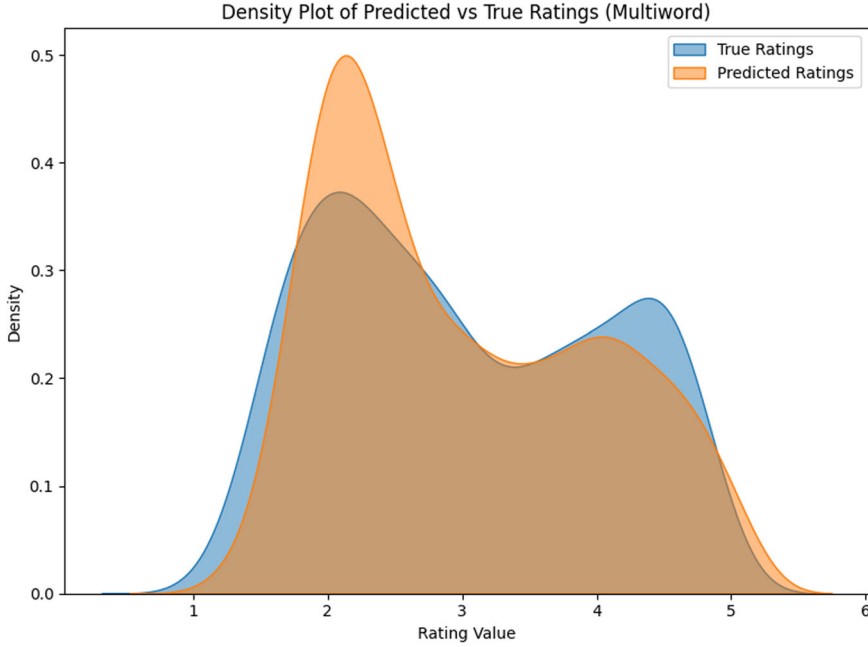

characteristic bimodal distribution (Figs. 2, 4), the Estonian dataset exhibits a different, more uniform distribution (Fig. 6), see also Fig. 1 in Proos and Aigro[50]. Despite being trained exclusively on bimodally-distributed English data, our model successfully generalizes to predict ratings in Estonian ($r = 0.68$ and $r = 0.80$ after post-hoc item exclusion), where concreteness follows a different distribution pattern. Yet still, it is clear from Fig. 7 that the model tries to predict bi-modally (i.e., the more "rugged" structure with ups and downs including the strong peak around a rating of 4.8). Nevertheless, the robust cross-lingual transfer, maintaining strong performance even with different underlying rating distributions, provides compelling evidence that our method may capture fundamental features of conceptual concreteness independent of any rating patterns specific to one language. The strong performance on both English single words ($r = 0.93$), multi-word expressions ($r = 0.85$), and Estonian ($r = 0.68$ and r = 0.80 after post-hoc item exclusion) demonstrates that our method provides reliable automated concreteness ratings that are in some important sense language-agnostic.

## Discussion

In this study, we introduced a, automated method for generating concreteness ratings that surpasses human-level reliability across multiple languages and expression types. By integrating multimodal transformers with emotion-finetuned language models, our approach achieves remarkable correlations with human ratings-$r = 0.93$ for English single words, $r = 0.85$ for English multi-word expressions, and $r = 0.68$ ($r = 0.80$ after post-hoc item exclusion) for Estonian expressions.

Our findings improve the accuracy of computational predictions of human concreteness ratings. Traditional methods, such as those based on static word embeddings or distributional statistics[25,33], have struggled to accurately predict concreteness ratings, particularly for abstract words and in cross-lingual contexts. By leveraging contextualized embeddings from transformers and incorporating visual and emotional information, our model captures nuanced semantic features critical for distinguishing between abstract and concrete concepts. The exceptional performance of our model relative to human inter-rater variability suggests that it captures underlying semantic properties more consistently than individual human judgments. This aligns with recent studies indicating that multimodal and emotion-aware models can enhance semantic representation[36].

Our approach also supports theories emphasizing the role of sensory and emotional experiences in concept representation[3,8]. These emphasize that lexical co-occurrence statistics alone may inadequately represent abstract concepts, as these concepts often rely on emotional and introspective associations rather than mere lexical context. Indeed, recent studies have demonstrated a significant interaction between concreteness, imageability, and emotion[51]. By integrating both emotional and visual dimensions, our multimodal, emotion-aware model inherently addresses this interaction, offering improved alignment with empirical evidence on how emotional valence influences concreteness perception.

The ability to automatically generate reliable concreteness ratings has significant implications for various domains, particularly in cognitive science and psycholinguistics. Traditionally, researchers in these fields have relied on manually collected concreteness ratings, which are time-consuming to obtain and often limited in scope due to practical constraints. Our model addresses this limitation by providing readily available concreteness measures for an extensive array of words and expressions, significantly expanding the resources available for empirical research. By applying these automated ratings to Estonian, a Uralic language outside the traditional WEIRD framework, our work broadens the linguistic diversity of concreteness research and equips researchers with tools to easily study cognitive processing in less frequently examined populations. Importantly, our approach overcomes a critical limitation of existing concreteness databases by enabling context-sensitive ratings. Rather than assigning fixed concreteness values to words in isolation, our model can process complete sentences and generate concreteness ratings that reflect how a word's concrete-abstract status shifts based on its surrounding context. The model generates a single concreteness rating for each provided individual word in a complete sentence. This context-sensitivity aligns with theoretical accounts emphasizing the dynamic, context-dependent nature of conceptual processing and provides researchers with a more ecologically valid tool for studying how humans process and represent meaning in natural language use[12].

In cognitive science, concreteness plays a crucial role in understanding how humans process and represent concepts. Concrete words, which refer to tangible objects or perceptible entities, are processed differently in the brain compared to abstract words, which denote intangible ideas or concepts. By providing accurate concreteness ratings, our model enables researchers to select and control stimuli more effectively in experimental designs, facilitating studies on semantic memory-the aspect of memory that involves the storage and retrieval of general world knowledge[6].

Moreover, in psycholinguistics, concreteness influences language acquisition, comprehension, and production. Children typically acquire

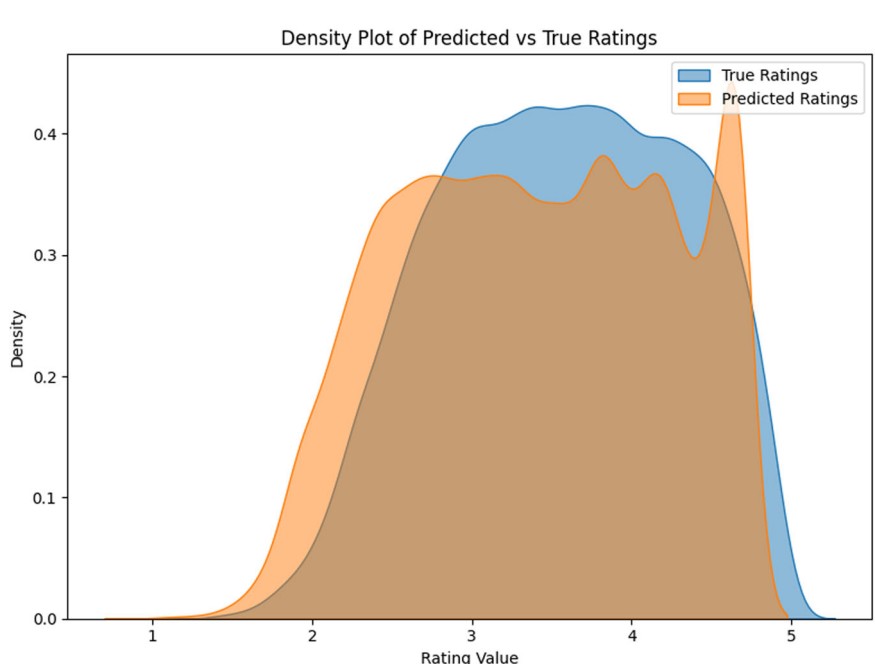

**Fig. 6 | Density heatmap of model predictions vs. true ratings for Estonian expressions after excluding post-hoc items with low agreement across participants ($r = 0.80$).** The main correlation score was $r = 0.68$.

**Fig. 7 |** Distribution comparison of all predicted vs. true ratings for Estonian expressions, showing successful capture of the rating distribution across languages.

concrete words earlier than abstract ones, and concrete language is often easier to process due to its direct connection to sensory experiences. Our model can aid in longitudinal studies examining language development by providing concreteness measures for vocabulary assessments, helping to identify patterns in how concreteness affects learning trajectories.

The neural correlates of abstract and concrete concepts are another area where our model can have a significant impact. Neuroimaging studies have shown distinct activation patterns in the brain when processing concrete versus abstract words, with concrete words engaging sensory and motor regions and abstract words involving more associative and linguistic areas[12,52]. By supplying precise and extensive concreteness ratings, our model facilitates the selection of stimuli for functional MRI or EEG studies, enhancing the reliability of findings related to the neural basis of language and concept representation. Furthermore, our model's ability to generate concreteness ratings across multiple languages opens up new avenues for cross-linguistic research. Psycholinguists can explore how concreteness influences bilingual language processing or compare the conceptual frameworks of different cultures. This could lead to insights into general versus language-specific aspects of cognition, informing theories about the interplay between language, thought, and perception.

In natural language processing (NLP), concreteness ratings are valuable features for enhancing the performance of various computational tasks. For instance, in metaphor detection, understanding the concreteness of words is essential because metaphors often involve describing abstract concepts using concrete terms[53]. By integrating our model's concreteness ratings, NLP systems can more effectively identify and interpret metaphorical language, improving applications like literary analysis, sentiment interpretation, and conversational agents. In sentiment analysis, concreteness can influence the emotional weight of words and phrases[54]. Concrete words might evoke stronger sensory or emotional responses, affecting the overall sentiment conveyed in a text. By accounting for concreteness, sentiment analysis algorithms can achieve better interpretations, distinguishing between literal and figurative language and improving the detection of sarcasm or irony. Machine translation is another area where our model can make a substantial contribution[55]. Translating abstract concepts accurately between languages requires a deep understanding of the semantic nuances that concreteness ratings can provide. Our method's robust cross-lingual performance suggests it can enhance translation quality by informing the selection of words that preserve the intended level of abstraction or concreteness in the target language. This is particularly valuable for idiomatic expressions or culturally specific concepts that are challenging to translate. Moreover, our model can significantly benefit NLP applications in multilingual contexts, especially for low-resource languages where annotated datasets are scarce. By automatically generating concreteness ratings, we can enrich linguistic resources without the need for extensive manual annotation, accelerating the development of NLP tools for these languages. This can improve information access and communication technologies in underrepresented linguistic communities, promoting digital inclusion.

Additionally, concreteness ratings can enhance text simplification algorithms by identifying complex abstract words that may be difficult for certain audiences, such as language learners or individuals with reading difficulties. By substituting these words with more concrete alternatives, we can make texts more accessible without altering the core content. In the realm of information retrieval and search engines, understanding the concreteness of query terms can refine search results. For example, queries containing abstract terms might benefit from results that provide definitions, explanations, or theoretical discussions, while concrete terms might lead to images, specifications, or tangible product information. This applicability extends to educational technologies. Vocabulary learning apps and language teaching platforms can use concreteness ratings to tailor content to learners' proficiency levels. Beginners might start with highly concrete words that are easier to visualize and remember, gradually progressing to more abstract vocabulary as their language skills develop. This personalized learning experience can enhance engagement and retention.

## Limitations
Our study has several limitations. First, it should be noted that the publicly available web-scale datasets used to train CLIP may contain inherent biases related to image-text pair distributions, potentially affecting semantic representation, especially for culturally specific or less represented concepts. Second, our model's predictions closely match the bimodal distribution characteristic of human concreteness ratings, but exhibit deviations in the intermediate range. This divergence likely reflects an inherent limitation of predictive regression models, which optimize predictions primarily for clearly concrete or abstract extremes. Intermediate concepts, inherently ambiguous and context-sensitive, are thus less reliably captured. Third, while the model generalizes well to Estonian, further evaluation is needed across a broader range of languages, especially those with different linguistic structures or from different language families. Languages with rich morphology, such as Finnish or Turkish, which use extensive inflection and agglutination, may present additional challenges for our model's tokenizer and embedding mechanisms. Similarly, languages with logographic writing systems like Mandarin Chinese or languages that use non-Latin scripts like Arabic and Hindi could affect the model's ability to generate accurate concreteness ratings due to differences in character representation and tokenization processes. Additionally, languages with high levels of homonymy or polysemy, or those that rely heavily on tonal distinctions, may introduce complexities that our model is not currently equipped to handle. Our model's multilingual applicability currently depends on translation modules rather than direct multilingual embedding spaces. Therefore, idiomatic expressions may not be fully captured. Simlarly, cultural context and linguistic nuances that influence concreteness perception might not transfer well through translation, leading to less reliable ratings. Future work should explore direct modeling in target languages without translation, possibly by training multilingual embeddings (e.g., using models such as mBERT, XLM-R, mGPT, XGLM, or mT5) or incorporating language-specific features, to enhance the general applicability of our method and ensure robust performance across diverse linguistic landscapes.

## Conclusion
Our method presents a significant advancement in automatically generating concreteness ratings, achieving beyond human-level reliability and demonstrating robust cross-lingual transfer. By integrating multimodal and emotion-aware embeddings, our model overcomes the limitations of previous approaches, providing a valuable tool for research and applications in cognitive science, psycholinguistics, and NLP. The findings underscore the potential of combining linguistic, visual, and emotional information to enhance semantic understanding in computational models.

## Data availability
To ensure that our tool is accessible to the research community and practitioners (even without coding skills), it is available as an easy-to-use web-based application under: concreteness.eu.

## Code availability
We have made a Github repository publicly available under https://github.com/ViktorKewenig/Automated_Concreteness, which includes comprehensive documentation, usage examples, and explanations on how to integrate the module into various applications.

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

## Acknowledgements
V.K. would like to thank Christopher Potts for creating NLP learning resources. This work was supported in part by the European Research Council Advanced Grant (ECOLANG, 743035); Royal Society Wolfson Research Merit Award (WRM370016) to G.V.; and Leverhulme award DS-2017-026 to V.K. and G.V.

## Author contributions
V.K. conceived the study, designed the methodology, implemented all analyses, curated the data, interpreted the results, created visualizations, and drafted the manuscript. J.I.S. critically reviewed and edited the manuscript for intellectual content. G.V. critically reviewed and edited the manuscript and secured the funding that supported this work.

## Competing interests
The authors declare no competing interests.
