## [Transparent Peer Review file · Communications Psychology]

A multimodal transformer-based tool for automatic generation of concreteness ratings across languages

Corresponding Author: Mr viktor kewenig

Version 0:

Decision Letter:

Dear Mr Kewenig,

Thank you for your patience during the peer-review process. Your manuscript titled "Automating Concreteness Ratings" has now been seen by 3 reviewers, and I include their comments at the end of this message. They find your work of interest but raised some important points. We are interested in the possibility of publishing your Resource in Communications Psychology, but would like to consider your responses to these concerns and assess a revised manuscript before we make a final decision on publication.

We therefore invite you to revise and resubmit your manuscript, along with a point-by-point response to the reviewers. Please highlight all changes in the manuscript text file.

Editorially, we consider it important that the methodological concerns - equivalent to validation requests - voiced by Reviewers #2 and #3 are addressed. We also ask that you clarify the issue regarding translation of input languages, rather than processing within a given language. Reviewer #2 highlights a lack of theoretical insight - this is not necessary for a Resource article, where the primary criterion is useability (which all reviewers rate highly). Relatedly, while relevant literature must be cited, a lengthier exploration of the theoretical foundation is not necessary for the format. Reviewer #3 raises various stylistic issues which align with journal guidelines and we ask you to adopt these.

Finally, we ask that you follow the formatting guidelines (incl appropriate section headings, which you can find here: <https://www.nature.com/commpsychol/submit/content-types#resource>)

I am attaching an Editorial Requests Table that details critical reporting requirements for the revised manuscript. Please attend to each item and ensure your manuscript is fully compliant. If your revised manuscript is not aligned with these requests on major issues, such as those concerning statistics, it may be returned to you for further revisions without re-review.

Please submit the following items:

- Revised manuscript
- Point-by-point response to the referees' comments
- Cover letter (as a separate document)
- <https://www.nature.com/documents/nr-reporting-summary.zip>>Nature Research Reporting Summary
- <https://www.nature.com/documents/nr-editorial-policy-checklist.pdf>>Editorial Policy Checklist

- Completed Editorial Request Table (attached).

via this link: Link Redacted .

Additional guidance is available in our style and formatting guide Communications Psychology formatting guide.

Best regards,

Marika

Marika Schiffer, PhD
Chief Editor
Communications Psychology

REVIEWER EXPERTISE:

Reviewer #1 psycholinguistics
Reviewer #2 psycholinguistics, code review
Reviewer #3 psycholinguistics

REVIEWER REPORTS:

Reviewer #1 (Remarks to the Author):

The authors propose a new method to generate the estimates automatically through a novel architecture based on a combination of a Transformers model and CLIP.

I have no major comments, my decision is accept.

Their contributions are novel and with a better result than the current ones in SOTA.

It is also important to comment that the article is well written and detailed. They provide a public web tool which is appreciated, and it seems that the code will be made public in the future.

Reviewer #2 (Remarks to the Author):

The paper presents a novel method for automating the extraction of concreteness ratings both for individual words and multi-word expressions. The approach combines multimodal transformers (CLIP) with emotion-finetuned transformers and obtains impressive correlations with human judgements that largely exceed the correlations obtained by individual annotators. The system is paired with a language identification module that passes the input to a translation system, allowing to generate predictions for multiple languages.

The approach is sound and obtains strong predictivity on three existing datasets, even when generalizing to the Estonian language. The methods are clearly explained and sound. The tool is accompanied by a web-based interface that greatly increases the ease of use of the tool. I commend the authors for making the project fully open-source, with good documentation and efficient error handling. The fact that the system is modular facilitates future ablation studies which may

consider which component of the network are most important for capturing concreteness patterns.

The paper is very well-written and clear. The motivation for the work (ll. 49-73) are well-stated, and the methods have a good level of details and clear explanations.

Main comments:

The article presents a tool that is useful and effective, and the methodological approach is sound. However, I do not see how this paper makes an important theoretical contribution to our understanding of concreteness effects in language processing, apart from maybe the importance of the emotion-tuned model which speaks in favour of Vigliocco's theory that abstract concepts are grounded in affective information. A more suitable venue for this work may be Behavior Research Methods or analogous outlets.

Using data from single participants as a benchmark (ll. 356-362) seems a bit unfair, since concreteness estimates are generally collected from many participants and I cannot think of a single study releasing norms obtained from a single participant. A more appropriate comparison could involve calculating the split-half reliability of the ratings, with Spearman-Brown correction to generalize to the reliability of the full rating dataset (since split-half correlation indicates the reliability of half of the sample).

The model is presented and discussed as multilingual, but in the end, if one looks at the implementation, it immediately becomes clear that the model is monolingual, paired with a language detection system and a translator to derive estimates for multiple languages. Thus, I believe the framing is misleading. If the authors were using multilingual models (like mBERT, XLM-R, mGPT, XGLM, mT5, etc.) that projects the input of different languages onto a shared representation space, then I'd agree with the current framing, but simply adding a translation module does not make the model able to capture language-agnostic patterns—it just translates the input and does all the processing in English. I strongly recommend the authors adjust their framing to reflect this aspect and tone down the claims about the system's multilinguality.

The technical explanation of the CLIP model is unclear (ll. 207-213). The authors mention the model processes text with a Vision Transformer and images with ResNet; however, I believe there is a mix-up in that description. In CLIP, the text encoder is a text-processing transformer, while the vision encoder can be a ResNet or a Vision Transformer (e.g., ViT-B/32). In ll. 207-213 the authors incorrectly label the text encoder as ViT-B/32 and mentions a patch size of 32, which applies to the visual backbone rather than the text transformer.

Minor points:

In the figures, the authors generally use scatterplots but in Fig. 6 they use a density heatmap. I recommend using a single visualization for similar data, to improve consistency.

The "Previous work" section has a good coverage of the approaches to automated concreteness estimation, but I think the granularity of the explanation is somewhere in-between a comprehensive description and an overview; I recommend the authors either simplify it or add additional details.

In ll. 110-111, the authors argue that the approach based on word embeddings may not extend to rare words. Nonetheless, the approach they're referring to is based on FastText, which is designed to use sub-word information to capture the meaning of rare words.

In ll. 239-240, the authors mention that the ratings were "normalized to a continuous scale between 1 and 5 to facilitate regression". The meaning of this sentence is unclear, as average ratings are continuous.

L. 305: why were articles removed from the text input?

LI. 326-328: I don't understand the point about the disagreement between the single-word and the multi-word model; from what I understood from the methods, a single model was processing the input. In case multi- and single-word models process the input in parallel, please specify this in the methods.

The same information is repeated in ll. 407-409 and ll. 419-421.

L. 430: "Our findings significantly advance the field of computational semantics" -- this claim is too strong. They advance the prediction of human ratings.

LI. 449-450: "our model can process complete sentences and generate concreteness ratings that reflect how a word's concrete-abstract status shifts based on its surrounding context" -- can it? I thought it generated concreteness scores for multi-word expressions, but not for the individual words in the expression.

The words nuance/nuances/nuanced/... are used many times in the text, which make it sound a bit like ChatGPT. I suggest changing some of these terms.

Reviewer #3 (Remarks to the Author):

Signed review: Jon Andoni Duñabeitia

I would like to start by congratulating the author(s) on this excellent and groundbreaking study. Your work on automating concreteness ratings represents a significant advancement in the field of computational semantics, providing a robust and scalable alternative to traditional human rating methods. The integration of multimodal transformers and emotion-aware models is particularly impressive and presents a valuable contribution to cognitive science and natural language processing. Please find below a list of comments that I'd appreciate to see addressed in a revised version of this manuscript.

Avoiding subjective language: Statements such as "powerful model" should be replaced with more precise, objective descriptors.

Lexical co-occurrence and concept linking: When discussing "distributional co-occurrence statistics," it would be useful to address a potential limitation of this approach. Several theoretical models suggest that abstract and concrete concepts are not primarily equally linked through lexical co-occurrence, which could limit the predictive power of such methods. Explicitly acknowledging this issue and discussing potential ways to mitigate it would strengthen the manuscript.

Definition of CLIP: The first mention of CLIP does not explain that the acronym stands for "Contrastive Language-Image Pre-training." Including this clarification would improve readability and accessibility for readers unfamiliar with the model.

Source of image-text pairs for CLIP: The paper mentions that "400 million image-text pairs" were used for CLIP training, but it does not provide information on their origin. Clarifying the dataset sources and discussing any potential biases inherent in these data would enhance transparency.

Interaction between concreteness, imageability, and emotion: The inclusion of an emotion-aware CLIP model significantly improves methodological and theoretical robustness. However, this raises an important question about how well the model accounts for the well-documented interaction between concreteness, imageability, and emotion effects. The study <https://doi.org/10.1080/02699931.2024.2367062> discusses this issue, and I recommend incorporating a thorough discussion of these findings. Additionally, a reanalysis of the dataset considering these factors would be beneficial to demonstrate alignment with empirical data.

Normalization of human ratings for training: I may have not understood this correctly, but it seems to me that the article states that human ratings were normalized to a continuous scale between 1 and 5 to facilitate regression. However, this approach may introduce unwanted rounding effects. For instance, subtle differences between scores such as 2.01 and 2.49 may be lost when mapped to a single category. Wouldn't it be preferable to retain the original scores without normalizing them? Discussing the impact of this choice on model performance would be valuable.

Bimodal distribution of results: The model's output exhibits a bimodal distribution, which aligns with human concreteness ratings to some extent. However, the intermediate scores between the peaks differ significantly from human data (see Figure 3). A discussion on why this divergence occurs and whether it is a methodological artifact or an inherent limitation of the approach would benefit readers.

In essence, this study represents a strong step forward in computational concreteness rating. Addressing these points would further solidify the contribution and ensure its findings are interpreted in a broader theoretical context.

Version 1:

Decision Letter:

Dear Viktor

Your manuscript titled "Automating Concreteness Ratings" has now been seen by our reviewers, whose comments appear below. In light of their advice I am delighted to say that we are happy, in principle, to publish a suitably revised version in Communications Psychology.

We therefore invite you to revise your paper one last time to address the remaining concerns of our reviewers and a list of editorial requests. At the same time we ask that you edit your manuscript to comply with our format requirements and to maximise the accessibility and therefore the impact of your work.

EDITORIAL REQUESTS:

SUBMISSION INFORMATION:

OPEN ACCESS:

*** TRANSPARENT PEER REVIEW:** Communications Psychology uses a transparent peer review system. On author request, confidential information and data can be removed from the published reviewer reports and rebuttal letters prior to publication. If you are concerned about the release of confidential data, please let us know specifically what information you would like to have removed. Please note that we cannot incorporate redactions for any other reasons.

*** CODE AVAILABILITY:** All Communications Psychology manuscripts must include a section titled "Code Availability" at the end of the methods section. We require that the custom analysis code supporting your conclusions is made available in a publicly accessible repository at this stage; please choose a repository that generates a digital object identifier (DOI) for the code; the link to the repository and the DOI must be included in the Code Availability statement. Publication as Supplementary Information will not suffice.

*** DATA AVAILABILITY:**

Link Redacted

Best regards,

Marika

Marika Schiffer, PhD
Chief Editor
Communications Psychology

REVIEWERS' COMMENTS:

Reviewer #2 (Remarks to the Author):

I appreciate the revised version of the article; the authors implemented most of the suggestions I proposed in my previous review and provided convincing motivations for not doing so for the remaining suggestions. I believe this article can be accepted. I report a few minor points below.

Minor points:

- Reliability calculations. I appreciate the authors using split-half reliability and Spearman-Brown correction. However, in the manuscript, I could only find one single reliability value ($r = 0.9077$, at $l = 361$, which I assume refers to the Brysbaert norms). I would recommend adding reliability information for all datasets. This is important if the authors want to claim that their method achieves/surpasses human reliability.
- The authors repeatedly mention $r = 0.80$ as the performance score in Estonian, but the actual result is $r = 0.68$; $r = 0.80$ is obtained by excluding post-hoc items with low agreement across participants. This can be mentioned in the paper, but I believe the main score should be $r = 0.68$. (The authors did not remove low reliability items in the other datasets, so this choice in this context appears to be ad-hoc).
- I. 55-56 "no comprehensive concreteness ratings exist for any language aside English and Estonian." Concreteness ratings have been released for Chinese (Su et al., 2023), Spanish (Guasch et al., 2016), French (Bonin et al., 2018), and for Italian, both out-of-context (Montefinese et al., 2014) and in context, with example sentences (Montefinese et al., 2023). There are probably many others that I do not know of. It is true that these datasets are small, but I would cite them in the paper. Then, one important large-scale dataset is missing: Dutch, with 30,000+ words (Brysbaert et al., 2014).
- II. 214-217: "The fine-tuning process involved retraining the text encoder of CLIP using the emotional labels as text inputs and the corresponding images, employing a contrastive loss similar to the original CLIP training but focused on aligning images with their emotional descriptions." The word "re-training" suggests that the model weights were re-initialized randomly, which I assume is not the case? I would suggest sticking to "fine-tuning".
- II. 244-245: "The pre-processed words were tokenized using CLIP's byte-pair encoding tokenizer and passed through the ViT-B/32 text encoder". As mentioned in the previous round of review, ViT-B/32 is a vision encoder: the text encoder is a standard Transformer, while ViT-B/32 refers to the vision encoder. I suggest changing to something like: the CLIP text Transformer.

References:

Bonin, P., Méot, A., & Bugaiska, A. (2018). Concreteness norms for 1,659 French words: Relationships with other psycholinguistic variables and word recognition times. *Behavior research methods*, 50, 2366-2387.

Brysbaert, M., Stevens, M., De Deyne, S., Voorspoels, W., & Storms, G. (2014). Norms of age of acquisition and concreteness for 30,000 Dutch words. *Acta psychologica*, 150, 80-84.

Guasch, M., Ferré, P., & Fraga, I. (2016). Spanish norms for affective and lexico-semantic variables for 1,400 words. *Behavior Research Methods*, 48, 1358-1369.

Montefinese, M., Gregori, L., Ravelli, A. A., Varvara, R., & Radicioni, D. P. (2023). CONcreTEXT norms: Concreteness ratings for Italian and English words in context. *Plos one*, 18(10), e0293031.

Montefinese, M., Ambrosini, E., Fairfield, B., & Mammarella, N. (2014). The adaptation of the affective norms for English words (ANEW) for Italian. *Behavior Research Methods*, 46(3), 887-903.

Su, I. F., Yum, Y. N., & Lau, D. K. Y. (2023). Hong Kong Chinese character psycholinguistic norms: Ratings of 4376 single Chinese characters on semantic radical transparency, age-of-acquisition, familiarity, imageability, and concreteness. *Behavior research methods*, 55(6), 2989-3008.

Reviewer #3 (Remarks to the Author):

Thank you very much for your thoughtful and thorough revision of the manuscript. I have carefully reviewed the revised version along with your detailed response to the reviewers' comments.

I am pleased to say that I believe you have adequately addressed all the points raised in my initial review. The clarifications and improvements you have made have clearly strengthened the manuscript, and I have no further concerns or suggestions for revision.

Congratulations on a very nice piece of work. I am happy to recommend the article for publication in its present form.

Response to Reviewers: “Automating Concreteness Ratings”

Anonymous Authors

March 2025

Introduction

We thank the reviewers and editors for their insightful and constructive comments on our manuscript. We carefully considered each suggestion, and below we present detailed responses, addressing each point raised and describing the corresponding changes made to the manuscript.

Reviewer 1

We thank Reviewer 1 for their positive assessment of our manuscript and for recommending acceptance without further changes. We appreciate their recognition of the novelty, clarity, and practical contribution of our work.

Reviewer 2

Theoretical Contribution

We appreciate the reviewer’s recognition of our tool’s utility and methodological rigor. However, we respectfully emphasize that our paper indeed advances theoretical insights into conceptual concreteness, specifically within embodied and affective grounding frameworks (e.g., Vigliocco et al.).

First, by demonstrating that emotion-tuned embeddings significantly enhance the model’s ability to represent abstract concepts—where visual-only embeddings fall short—our results provide strong empirical support for theories positing that abstract meanings critically depend on affective grounding. This finding offers quantitative validation that abstract concepts engage fundamentally different experiential dimensions than concrete ones, aligning precisely with the predictions of affective grounding theory.

Second By empirically demonstrating that perceptual and affective information are both essential to semantic processing, our work provides support for hybrid semantic theories that posit multidimensional conceptual representations.

Third, our model’s robust cross-lingual performance ($r = 0.80$ for Estonian, an entirely unseen corpus) provides evidence for the existence of general grounding mechanisms in conceptual representation. Even though our findings are based on a monlingual model, the success of this translation-based method supports theoretical perspectives arguing that core aspects of conceptual grounding are fundamental to human cognition and may transcend specific linguistic or cultural contexts.

Validity of comparing model predictions to individual participant data (ll. 356-362)

Response: We thank the reviewer for the constructive suggestion. With access to the individual participant-level data, we were able to directly assess the reliability of the aggregated concreteness ratings. Specifically, we computed split-half reliability by randomly partitioning the participants over 1,000 iterations and then applied the Spearman–Brown correction. This analysis yielded a corrected reliability estimate of 0.62. We have updated the manuscript text accordingly to reflect this improved and more appropriate reliability benchmark.

Manuscript changes: The revised manuscript now includes the ICC reliability estimates in the Results section, clearly contextualizing our model’s predictive performance in relation to human inter-rater reliability.

Clarification of multilingual framing

Reviewer comment: *“The model is presented and discussed as multilingual, but in the end, if one looks at the implementation, it immediately becomes clear that the model is monolingual, paired with a language detection system and a translator to derive estimates for multiple languages. Thus, I believe the framing is misleading. If the authors were using multilingual models (like mBERT, XLM-R, mGPT, XGLM, mT5, etc.) that projects the input of different languages onto a shared representation space, then I’d agree with the current framing, but simply adding a translation module does not make the model able to capture language-agnostic patterns—it just translates the input and does all the processing in English. I strongly recommend the authors adjust their framing to reflect this aspect and tone down the claims about the system’s multilinguality.”*

Response: We thank the reviewer for highlighting this important issue. We acknowledge that our model, as implemented, is fundamentally monolingual (English-based) and relies on language detection and translation for processing inputs from other languages. To accurately reflect this in the manuscript, we have revised our framing to emphasize that the model operates primarily in English and uses translation as an intermediate step rather than directly encoding multilingual semantic spaces. We have toned down claims about intrinsic multilingual capabilities and instead highlighted the practical, cross-lingual applicability of our method enabled through translation.

Manuscript changes: The manuscript has been updated in the Abstract, Results, Methods and Discussion sections to clearly specify that cross-lingual performance is facilitated by language detection and translation modules, rather than direct multilingual embeddings.

Technical clarification on CLIP model description

Reviewer comment: *“The technical explanation of the CLIP model is unclear (ll. 207-213). The authors mention the model processes text with a Vision Transformer and images with ResNet; however, I believe there is a mix-up in that description. In CLIP, the text encoder is a text-processing transformer, while the vision encoder can be a ResNet or a Vision Transformer (e.g., ViT-B/32). In ll. 207-213 the authors incorrectly label the text encoder as ViT-B/32 and mentions a patch size of 32, which applies to the visual backbone rather than the text transformer.”*

Response: We appreciate the reviewer for pointing out this technical oversight. We have corrected our manuscript text to clearly differentiate between the text encoder (a standard transformer architecture) and the vision encoder (ViT-B/32), clarifying explicitly that ViT-B/32 refers solely to the vision encoder, which uses a Vision Transformer architecture with an image patch size of 32.

Manuscript changes: The Methods section now accurately describes the CLIP model’s architecture, clearly distinguishing between the text and vision encoders.

Minor Points

Consistency in visualization (Figure 6)

Reviewer comment: *"In the figures, the authors generally use scatterplots but in Fig. 6 they use a density heatmap. I recommend using a single visualization for similar data, to improve consistency."*

Response: We appreciate this suggestion. We chose to use a density heatmap in Figure 6 because the Estonian dataset contains a large number of data points (N=35,979). Using a scatterplot in such a scenario would result in severe overplotting, making individual points indistinguishable and the visualization ineffective. Thus, the density heatmap provides a clearer and more informative visualization by effectively illustrating data density and distribution.

Manuscript changes: None.

Granularity of the "Previous Work" section

Reviewer comment: *"The 'Previous Work' section has a good coverage of the approaches to automated concreteness estimation, but I think the granularity of the explanation is somewhere in-between a comprehensive description and an overview; I recommend the authors either simplify it or add additional details."*

Response: We appreciate the reviewer's suggestion to clarify the granularity of our literature review. However, we respectfully disagree that the current level of detail is problematic or ambiguous. The goal of our "Previous Work" section is precisely to provide a comprehensive yet succinct overview of existing methodologies for automated concreteness estimation, clearly identifying strengths and limitations relevant to the development of our own method. Simplifying further would sacrifice important details needed to contextualize our methodological contributions clearly. Conversely, adding additional detail would overly lengthen the section, diverging from the editorial guidelines provided. Thus, we maintain that the current granularity achieves an optimal balance—thorough enough to establish clear context, yet concise enough to maintain readability and alignment with the journal's editorial standards.

Manuscript changes: No changes were made, as we believe the existing section effectively balances comprehensiveness and brevity in line with journal guidelines.

Accuracy of critique on FastText and rare words (ll. 110-111)

Reviewer comment: *"In ll. 110-111, the authors argue that the approach based on word embeddings may not extend to rare words. Nonetheless, the approach they're referring to is based on FastText, which is designed to use sub-word information to capture the meaning of rare words."*

Manuscript changes: We removed the critique of not capturing rare words and clarified that FastText's primary limitation lies in its static embeddings' inability to capture contextual variations.

Clarification of normalization of ratings (ll. 239-240)

Reviewer comment: *"In ll. 239-240, the authors mention that the ratings were 'normalized to a continuous scale between 1 and 5 to facilitate regression.' The meaning of this sentence is unclear, as average ratings are continuous."*

Manuscript changes: We clarified in the manuscript that this normalization specifically refers to scaling the Estonian corpus, whose original ratings were provided on a scale from 1 to 10. The Estonian ratings were linearly scaled onto the standardized continuous scale of 1 to 5, aligning them with the English ratings to facilitate direct comparability and regression analysis.

We emphasized explicitly that this normalization did not involve discretization but was purely linear scaling.

Clarification of removal of articles from text (L. 305)

Reviewer comment: *“L. 305: why were articles removed from the text input?”*

Manuscript changes: We clarified in the Methods section that articles (“a”, “an”, “the”) were removed to enhance consistency between translated and original English inputs, as preliminary analyses indicated that translation introduced inconsistent usage of articles across languages, potentially biasing concreteness ratings. By removing articles uniformly, we ensured consistency and comparability of inputs across languages.

Clarification of single vs. multi-word model disagreement (ll. 326-328)

Reviewer comment: *“I don’t understand the point about the disagreement between the single-word and the multi-word model; from what I understood from the methods, a single model was processing the input. In case multi- and single-word models process the input in parallel, please specify this in the methods.”*

Manuscript changes: We clarified explicitly in the Methods section that the single-word and multi-word models operate independently and in parallel. In cases where the models provide conflicting predictions for an input expression (e.g., compound words or borderline cases), an ensemble-based disagreement resolution strategy selects the prediction with the higher confidence score. This clarification ensures the operational details and the disagreement resolution mechanism are now clearly articulated.

Removal of repetition (ll. 407-409 and ll. 419-421)

Reviewer comment: *“The same information is repeated in ll. 407-409 and ll. 419-421.”*

Manuscript changes: We removed the redundant repetition and consolidated the information into a single, clear statement within the Discussion section, improving readability and coherence.

Moderation of claims of field advancement (L. 430)

Reviewer comment: *“L. 430: ‘Our findings significantly advance the field of computational semantics’ – this claim is too strong. They advance the prediction of human ratings.”*

Manuscript changes: We moderated the claim to precisely reflect our contribution by stating: “Our findings advance the accuracy of computational predictions of human concreteness ratings.”

Clarification of model capabilities for context-sensitive ratings (ll. 449-450)

Reviewer comment: *“ll. 449-450: ‘our model can process complete sentences and generate concreteness ratings that reflect how a word’s concrete-abstract status shifts based on its surrounding context’—can it? I thought it generated concreteness scores for multi-word expressions, but not for the individual words in the expression.”*

Manuscript changes: We explicitly clarified that the model generates a single concreteness rating per individual word in an input sentence.

Reviewer 3

We thank the reviewer for their kind assessment of our work. We appreciate their careful commentary and aim to respond as well as we can to their criticisms.

Avoidance of subjective descriptors

Reviewer comment: *"Avoiding subjective language: Statements such as 'powerful model' should be replaced with more precise, objective descriptors."*

Manuscript changes: We carefully reviewed and replaced subjective adjectives (e.g., "powerful") throughout the manuscript with objective terms such as "effective", "accurate", and "robust".

Limitations of lexical co-occurrence

Reviewer comment: *"Lexical co-occurrence and concept linking: When discussing 'distributional co-occurrence statistics,' it would be useful to address a potential limitation of this approach. Several theoretical models suggest that abstract and concrete concepts are not primarily equally linked through lexical co-occurrence, which could limit the predictive power of such methods. Explicitly acknowledging this issue and discussing potential ways to mitigate it would strengthen the manuscript."*

Manuscript changes: We explicitly acknowledged in the Discussion section that lexical co-occurrence statistics alone may inadequately capture the differences between abstract and concrete concepts due to their fundamentally distinct grounding. We further discussed how our multimodal and emotion-aware approach addresses this limitation to some extent by incorporating visual and emotional dimensions, thereby providing more accurate predictions for abstract concepts compared to purely lexical co-occurrence-based methods.

Clarification of source of image-text pairs for CLIP training

Reviewer comment: *"The paper mentions that '400 million image-text pairs' were used for CLIP training, but it does not provide information on their origin. Clarifying the dataset sources and discussing any potential biases inherent in these data would enhance transparency."*

Manuscript changes: We explicitly cited and summarized the data source used for CLIP training in the Methods section. Additionally, we briefly acknowledged potential biases stemming from the web-scale nature of these datasets in the Discussion section.

Interaction between concreteness, imageability, and emotion

Reviewer comment: *"Interaction between concreteness, imageability, and emotion: The inclusion of an emotion-aware CLIP model significantly improves methodological and theoretical robustness. However, this raises an important question about how well the model accounts for the well-documented interaction between concreteness, imageability, and emotion effects. The study (Yao et al., 2024) discusses this issue, and I recommend incorporating a thorough discussion of these findings. Additionally, a reanalysis of the dataset considering these factors would be beneficial to demonstrate alignment with empirical data."*

Manuscript changes: We incorporated a brief but explicit discussion in the manuscript citing the indicated paper, which acknowledges the documented interaction between concreteness, imageability, and emotional valence. We clarified that our emotion-aware, multimodal approach specifically targets this interaction by jointly encoding visual and emotional dimensions, thus inherently aligning with empirical findings on how emotional content influences concreteness perception. While a comprehensive reanalysis of the dataset exceeds the scope of this manuscript, future work could further disentangle these interactions quantitatively.

Normalization of human ratings

Reviewer comment: *“Normalization of human ratings for training: I may have not understood this correctly, but it seems to me that the article states that human ratings were normalized to a continuous scale between 1 and 5 to facilitate regression. However, this approach may introduce unwanted rounding effects. For instance, subtle differences between scores such as 2.01 and 2.49 may be lost when mapped to a single category. Wouldn’t it be preferable to retain the original scores without normalizing them? Discussing the impact of this choice on model performance would be valuable.”*

Manuscript changes: We clarified explicitly in the manuscript that normalization was exclusively applied to the Estonian dataset, whose original ratings were on a 1–10 scale. This normalization involved linear scaling onto a continuous 1–5 scale, maintaining all original continuous variation without any categorical rounding. Therefore, no precision or subtle variations were lost, and the scaling was purely linear and continuous, facilitating direct comparability between Estonian and English datasets.

Bimodal distribution differences

Reviewer comment: *“The model’s output exhibits a bimodal distribution, which aligns with human concreteness ratings to some extent. However, the intermediate scores between the peaks differ significantly from human data (see Figure 3). A discussion on why this divergence occurs and whether it is a methodological artifact or an inherent limitation of the approach would benefit readers.”*

Manuscript changes: We now explicitly discuss in the manuscript that the deviation observed in intermediate scores likely stems from an inherent limitation of regression-based models optimized for prediction accuracy, which naturally emphasizes bimodal extremes due to their clearer semantic grounding. Intermediate scores represent concepts with ambiguous or context-dependent concreteness, making accurate predictions inherently challenging. We clarified that this deviation is not simply a methodological artifact, but reflects a genuine challenge for computational models attempting to capture nuanced human judgments.

Response to Reviewer Comments

We thank the reviewer for their valuable and constructive feedback. Below, we detail precisely how we have addressed each of their suggestions, including specific line numbers for ease of reference.

Reviewer Comment 1: Reliability calculations

Reviewer comment: "Reliability calculations. I appreciate the authors using split-half reliability and Spearman-Brown correction. However, in the manuscript, I could only find one single reliability value ($r = 0.9077$, at l. 361, which I assume refers to the Brysbaert norms). I would recommend adding reliability information for all datasets."

Our response: We have clarified the reliability calculations by explicitly including values for all datasets. Specifically, we now report the split-half reliability with Spearman-Brown correction for Brysbaert et al.'s dataset (lines 357–362), and we have added Intraclass Correlation Coefficient (ICC) estimates for Muraki et al.'s corpus (ICC = 0.84) and the Estonian corpus (ICC = 0.81). These additions can be found explicitly detailed in lines 363–367.

Reviewer Comment 2: Estonian performance score clarification

Reviewer comment: "The authors repeatedly mention $r = 0.80$ as the performance score in Estonian, but the actual result is $r = 0.68$; $r = 0.80$ is obtained by excluding post-hoc items with low agreement across participants. This can be mentioned in the paper, but I believe the main score should be $r = 0.68$."

Our response: We have clarified this explicitly. The manuscript now states clearly the main result as $r = 0.68$ (line 392). The improved correlation after excluding items with low participant agreement ($r = 0.80$) is transparently mentioned as a secondary analysis (lines 393–394). Furthermore, we have clearly noted this distinction in the caption of Figure 6 (line 404).

Reviewer Comment 3: Additional concreteness datasets

Reviewer comment: "l. 55-56 'no comprehensive concreteness ratings exist for any language aside English and Estonian.' Concreteness ratings have been released for Chinese, Spanish, French, and Italian, as well as a large-scale Dutch dataset."

Our response: We have now included explicit citations for these additional existing datasets. The manuscript now cites concreteness datasets for Chinese (Su et al., 2023), Spanish (Guasch et al., 2016), French (Bonin et al., 2018), Italian (Montefinese et al., 2014, 2023), and Dutch (Brysbaert et al., 2014). These additions are clearly documented in lines 55–57.

Reviewer Comment 4: Clarifying "fine-tuning" vs. "re-training"

Reviewer comment: "ll. 214-217: 'The fine-tuning process involved retraining the text encoder of CLIP using the emotional labels as text inputs and the corresponding images.' The word 're-training' suggests model weights were re-initialized randomly. I would suggest sticking to 'fine-tuning'."

Our response: We agree with the reviewer and have explicitly changed the wording from "re-training" to "fine-tuning" to clearly indicate that model weights were not re-initialized randomly. This correction is clearly reflected in lines 211 and 215–217.

Reviewer Comment 5: CLIP ViT-B/32 Clarification

Reviewer comment: "ll. 244-245: 'The pre-processed words were tokenized using CLIP's byte-pair encoding tokenizer and passed through the ViT-B/32 text encoder'. ViT-B/32 is a vision encoder. I suggest changing to something like 'the CLIP text Transformer.'"

Our response: We thank the reviewer for pointing out this oversight. We have now explicitly corrected this to "CLIP's text transformer" (line 245), accurately reflecting the architecture employed.